# Analysis and Kinetics Modeling of the Isothermal Oxidation Behavior of Silicide Coatings

Dongyang An [1,2,*], Jingsheng Zhang [3], Zhipeng Liang [4], Yunji Xie [4], Mingyu Gao [4], Deshun Sun [5], Peng Xiao [2] and Jingmin Dai [2]

1   Beijing Xinghang Mechanical-Electrical Equipment Co., Ltd., Beijing 100074, China
2   School of Instrumental Science and Engineering, Harbin Institute of Technology, Harbin 150001, China
3   National Key Laboratory of Electromagnetic Space Security, Tianjin 300000, China
4   School of Mechatronics Engineering, Harbin Institute of Technology, Harbin 150001, China; gmy0523@foxmail.com (M.G.)
5   Intelligent Medical Innovation institute, Southern University of Science and Technology Hospital, Shenzhen 518035, China
*   Correspondence: zdady1989@163.com

**Abstract:** In this paper, an online apparatus was developed for isothermal thermogravimetric measurement of silicide coatings within a wide temperature range (from −180 °C to 2300 °C) based on thermogravimetric analysis. Firstly, the measuring principle and method regarding silicide coatings of this apparatus were studied. Secondly, on the basis of oxidation kinetics analysis, the intrinsic mechanism and kinetic parameters of three stages (oxidation, diffusion, and fall-off) of silicide coatings were studied, and the oxidation kinetics features were also analyzed. In addition, according to mathematical physics methods, a kinetics model of silicide coatings in different stages of oxidation was established, including parameters such as weight change, oxidation rate, oxidation time, etc. Finally, online isothermal experiments from −180 °C to 2300 °C were carried out and analyzed. The results showed that the kinetic model established in this paper was in good agreement with the oxidation process of silicide coatings. In this paper, a complete kinetics model including different oxidation stages is proposed for the entire oxidation process of a silicide coating, revealing its oxidation mechanism. The research will play a significant role in the study of preparation technology improvement and high-temperature environment application. This paper studied two measuring methods: weight gain and weight loss measuring methods. Also, an experiment was carried out on the silicide coatings to explore the physical oxidation process between −180 °C and 2300 °C. The results proved the perfect consistency of the kinetics model proposed by this paper and the oxidation process of silicide coatings. This paper will play a significant role in the study of preparation technology enhancement and high-temperature environment application. It also provides a theoretical foundation for accelerated aging and life evaluation methods.

**Keywords:** thermogravimetric analysis; measurement; kinetics analysis; intrinsic mechanism; silicide coatings

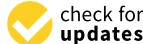



## 1. Introduction

As one of the most important and mature high-temperature-protective coatings, silicide coatings are extensively applied in fields such as the aviation industry, space industry, military affairs, and the civilian industry [1–3]. Early in 1965, the Sylvania Company manufactured silicide coatings and successfully applied them to the Nb alloy nozzle of a Pratt & Whitney F100 rocket engine [4]. The protective effect of silicide on substrates refers to the dense $SiO_2$ coating formed by oxidation behavior on the surface, which can not only prevent oxygen from diffusing to substrates, but also remedy defects such as cracks and holes generated in the process of oxidation [5,6]. The formation of $SiO_2$ and the loss of Si happen simultaneously during the usage of silicide coatings. When the generated

$SiO_2$ is not sufficient to form a completely dense coating or cannot remedy the increasing defects, oxygen will diffuse to the substrates, causing oxidation and failure [7]. Therefore, oxidation behavior is the key element that causes the failure of silicide coatings. Revealing the oxidation mechanism plays a significant role in the study of improving preparation technology and high-temperature environment application.

With the accumulation of a large amount of experimental data, oxidation theory continues to develop. Lowell [8] established an oxidation model based on the cracking of oxide layers. Smialek [9] developed another oxidation model according to the statistical rule of fall-off. Both models could be applied to analyze the process of oxidation kinetics. Chan [10–14] revealed the mechanism of the destruction of oxide layers in theory. The fracture mechanics theory was employed to describe the fall-off and breakdown phenomenon of oxide layers and a model describing material cyclic oxidation behavior was established. Lei [15] studied the bending and fall-off process of oxide scales according to the buckling theory of elastic beams. In addition, for the kinetics research of material oxidation, the first-order reaction and multi-stage reaction were used to describe the oxidation behavior [16–20]. Due to the complexity of the breakdown modes [21–24], different theories have distinct scopes of application. Hence, different theoretical models should be established for different materials. Oxidation has a distinct impact on properties, but the existingresearch has rarely proposed a complete model of oxidation kinetics for the entire oxidation process.

Based on thermogravimetric analysis, an online apparatus for isothermal thermogravimetric measurement was developed to achieve online isothermal thermogravimetric experiments of silicide coatings from −180 °C to 2300 °C. Secondly, the measuring method for silicide coatings was studied via this apparatus. In addition, based on oxidation kinetics analysis, in this paper, the intrinsic mechanism and the kinetics parameters of different oxidation stages of silicide coatings were studied and their oxidation kinetics features were analyzed. Finally, according to mathematical physics methods, a kinetics model of silicide coatings in different stages of oxidation was established. Experimental verification and analysis were also conducted.

## 2. Thermogravimetric Experimental Apparatus

Based on thermogravimetric analysis, an experimental thermogravimetric apparatus of isothermal oxidation for silicide coatings was developed, which could conduct an online isothermal thermogravimetric experiment of Nb-Hf alloy substrates coated with silicide materials in a wide temperature range (−180 °C~2300 °C). The precision of the online weighing system was ±0.1 mg. The apparatus mainly contained a high-temperature heating furnace (1700 °C~2300 °C), moderate-temperature heating furnace (100 °C~1800 °C), room-temperature heating furnace (20 °C~120 °C), low-temperature heating/cooling furnace (−180 °C~20 °C), high-temperature-resistant chuck, specimen moving system (horizontal moving rail and vertical moving rail), real-time weighing system, cooling water system, control system, and computer data acquisition system. The structure diagram and actual figure of the apparatus are shown in Figure 1. In the experiment, the specimen moving system moved the specimen to put it into different furnaces for heating or cooling. The real-time temperature signal was transmitted to the control cabinet for display, storage, analysis, and calculation. The control cabinet sent control messages through cables to the temperature control device of each furnace and the transmission components for specimen moving.

The high-temperature heating furnace heated the specimen through induction heating and instantly measured its temperature through fiber-optic pyrometer (SMART-FG-3522). The moderate-temperature heating furnace used a spiral graphite electrode to radiantly heat the specimen and measured its temperature through type K thermocouple in real time. Room-temperature heating furnace radiantly heated the specimen by using a resistive film. A PT100 platinum resistance thermometer was used to measure the specimen in real time. Low-temperature heating/cooling furnace used a resistance wire to radiantly heat the specimen while cooling it with liquid nitrogen, using PT100 platinum resistance ther-

mometer to instantly measure the temperature of the specimen.The four heating/cooling furnaces and the supporting components of the specimen moving system were placed on the same test bench and installed in parallel. Driven by the horizontal moving rail, the fixture positioning device and the high-precision weighing balance moved horizontally in the direction parallel to the four furnaces, which enabled the specimen to move between different furnaces. The vertical moving rail drove the weighing balance to pull the fixture positioning device so that they could move up and down in the direction perpendicular to the furnaces, which enabled the in-putting and out-taking of the specimen from different furnaces. The laser focus positioning method was used by the specimen moving system. A laser positioning transmitter was installed vertically under the specimen moving arm, while a laser positioning receiver was installed beside the heating furnaces. After the receiver received the signals from the transmitter, the specimen moving was stopped. The opening on the top of each furnace was quite small, which enabled the specimen moving system to both meet the temperature measuring requirements and ensure a better heating/cooling effect on the specimen.

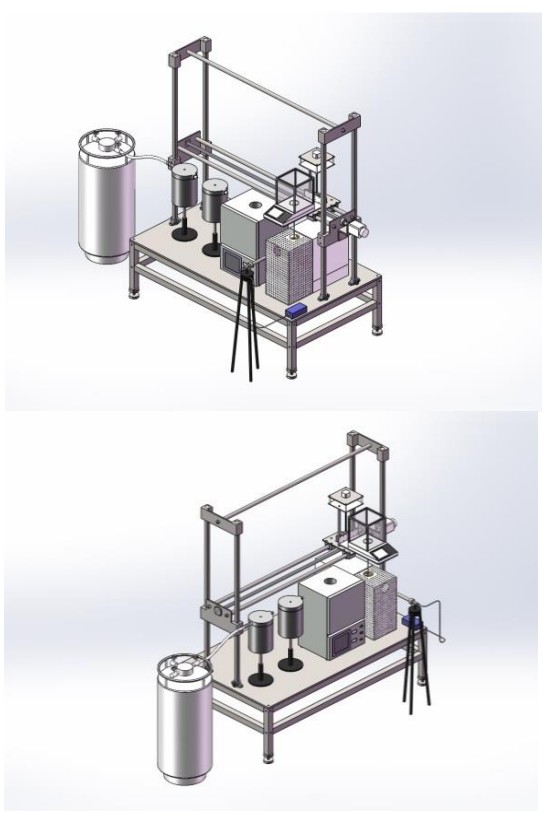

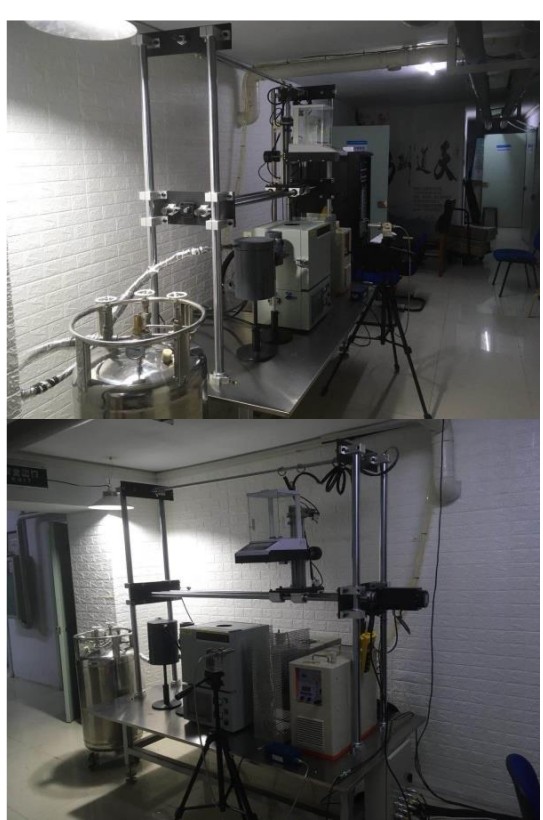

(**a**) Structure diagram.                                                                 (**b**) Actual figure.

**Figure 1.** Thermogravimetric experimental apparatus for isothermal oxidation. High-temperature heating furnace, moderate-temperature heating furnace, room-temperature heating furnace, and low-temperature heating/cooling furnace were the core components.

## 3. Measuring Principle and Methods

The thermogravimetric experiment under isothermal conditions mainly included a weight gain experiment and weight loss experiment, which means that a high-precision weighing balance was required to obtain the weight of the coating at different temperatures or times. According to the surface area of the specimen, the oxidation weight change per unit area and the average oxidation rate of the coating within a certain period of time were obtained. Finally, the oxidation resistance of the coating was analyzed.

### 3.1. Weight Gain Measuring Method

The weight gain measuring method is better applied to coating materials less prone to producing volatile oxides. The experimental temperature is set according to the experimental requirements. A high-precision weighing balance is used to obtain the weight of the specimen $m_{h1}$(g) before the experiment. After $t_{h1}$ min, the weight of the specimen is $m_{h2}$(g). $S_{h1}$ is the original surface area (m$^2$).The oxidation weight gain per unit is $G_h{}^+$(g/m$^2$):

$$G_h{}^+ = \frac{m_{h2} - m_{h1}}{S_{h1}} \tag{1}$$

The average oxidation rate is $\overline{K_h{}^+}$(g/m$^2 \cdot$ min):

$$\overline{K_h{}^+} = \frac{\overline{G_h{}^+}}{t_{h1}} \tag{2}$$

### 3.2. Weight Loss Measuring Method

The weight loss measuring method is applicable to coating materials which produce volatile oxides under experimental temperatures. The experimental temperature is set according to the experimental requirements. A high-precision weighing balance is used to obtain the weight of the specimen $m_{h3}$(g) before the experiment. After $t_{h2}$ min, the weight of the specimen is $m_{h4}$(g). $S_{h2}$ is the original surface area (m$^2$).

The oxidation weight loss per unit is $G_h{}^-$(g/m$^2$):

$$G_h{}^- = \frac{m_{h3} - m_{h4}}{S_{h2}} \tag{3}$$

The average oxidation rate is $\overline{K_h{}^-}$(g/m$^2 \cdot$ min):

$$\overline{K_h{}^-} = \frac{\overline{G_h{}^-}}{t_{h2}} \tag{4}$$

## 4. Oxidation Kinetics Model of Silicide Coatings

According to the kinetics principle, the steps of oxidation behavior of silicide coatings are as follows [25–28]: (1) the oxidation reaction on the boundary, (2) the inner diffusion of oxygen through the product layer, (3) the outer diffusion of silicon through the product layer, (4) fall-off of the reaction products. Thus, the isothermal oxidation kinetics of silicide coatings can be divided into three stages. The first stage of the oxidation reaction: due to the pretty thin reaction product, the entire oxidation rate is controlled by the boundary chemical reaction. In the second stage of the oxidation reaction, with a thicker product layer, a longer diffusion path of oxygen through the product layer, and increasing resistance, the oxidation rate is mainly affected by diffusion. In the third stage of the oxidation reaction, the surface of the product layer continues to fall off with chemical reaction, diffusion, and fall-off. The oxidation rate is mainly influenced by fall-off.

In this study, the specimen used a Nb-Hf substrate coated by silicide. The major component of the silicide coating was $NbSi_2$, whose oxidation product was $Nb_5Si_3$. The sectional area of the coating is $S_D$. The density of $NbSi_2$ and $Nb_5Si_3$ are $\rho_{NbSi_2}$ and $\rho_{Nb_5Si_3}$, respectively. The oxidation thickness of $NbSi_2$ is $x$. During the oxidation process, the rate of mass change with time of NbSi$_2$ with oxidation thickness x, namely, the oxidation rate of NbSi$_2$ per unit time, is as follows:

$$v_{D1} = \frac{\mathrm{d}\left[S_D x \rho_{NbSi_2}\right]}{\mathrm{d}t} = S_D \rho_{NbSi_2} \frac{\mathrm{d}x}{\mathrm{d}t} \tag{5}$$

It is assumed that the mass of $NbSi_2$ with thickness $x$ is $W_0$; when $Nb_5Si_3$ is completely oxidized, the mass of the sample becomes $W_1$, and the change in the oxidation mass of

the sample is $\Delta W$. In the oxidation process, the weight gain of the specimen per unit area is $\frac{\Delta W}{S_D}$:

$$\frac{\Delta W}{S_D} = \frac{W_1 - W_0}{S_D} = \frac{\alpha x S_D \rho_{Nb_5Si_3} - x S_D \rho_{NbSi_2}}{S_D} \tag{6}$$

In Equation (6), $\alpha$ is the thickness conversion factor, which weights $x \cdot \rho_{NbSi_2} \cdot S_D$, and produces $\rho_{NbSi_2}$ weighted $\alpha \cdot x \cdot \rho_{Nb_5Si_3} \cdot S_D$ in the process of oxidation. The oxidation thickness of the specimen is $x$:

$$x = \frac{1}{a\rho_{Nb_5Si_3} - \rho_{NbSi_2}} \cdot \frac{\Delta W}{S_D} \tag{7}$$

(1)    First Stage of Oxidation

The surface of the coating was directly exposed to oxygen. With the increase in temperature, the coating reacted with oxygen. At this time, the oxidation rate of NbSi$_2$ ($v_{D1}$) equals the chemical reaction rate $v_h$, and

$$v_h = S_D \cdot k_1 \cdot C \tag{8}$$

In Equation (8), $k_1$ is the oxidation rate constant of the chemical reaction between oxygen and NbSi$_2$. $C$ is the oxygen concentration in the air. According to Equations (5) and (8), we have:

$$\frac{d\left[S_D \cdot x \cdot \rho_{NbSi_2}\right]}{dt} = S_D \cdot \rho_{NbSi_2} \frac{dx}{dt} = S_D \cdot k_1 \cdot C \tag{9}$$

The integral of Equation (9) is as follows:

$$\int_0^x dx = \int_0^t \frac{k_1 \cdot C}{\rho_{NbSi_2}} dt \tag{10}$$

Therefore,

$$x = \frac{k_1 \cdot C}{\rho_{NbSi_2}} t \tag{11}$$

According to Equations (7) and (11),

$$\Delta W_1 = \frac{S_D \cdot k_1 \cdot C\left[a\rho_{Nb_5Si_3} - \rho_{NbSi_2}\right]}{\rho_{NbSi_2}} t \tag{12}$$

In Equation (12), $\Delta W_1$ is the mass change during oxidation. The kinetics model of the first stage of oxidation is.

$$\Delta W_1 = K_y t \tag{13}$$

where $K_y = \frac{S_D \cdot k_1 \cdot C\left[a\rho_{Nb_5Si_3} - \rho_{NbSi_2}\right]}{\rho_{NbSi_2}}$.

Equation (13) shows that at the first stage of oxidation, the oxidation weight gain and oxidation time of *NbSi$_2$* have a linear relationship.

(2)    Second Stage of Oxidation

After the oxidation reaction continued for a while, the oxidation product had a certain thickness. The diffusion path of oxygen through the oxidation products was lengthened and the diffusion resistance increased. At this time, diffusion became the limitation of the entire oxidation reaction. The whole oxidation reaction rate can be represented by diffusion rate $v_{D1}$ can be represented by diffusion rate $v_k$.

$$v_k = S_D \cdot C \cdot \frac{D}{x} \tag{14}$$

In this equation, $D$ is the diffusion coefficient of oxygen in the oxide scale.

According to Equations (5) and (14), we obtain

$$\frac{d\left[S_D \cdot x \cdot \rho_{NbSi_2}\right]}{dt} = S_D \cdot \rho_{NbSi_2} \frac{dx}{dt} = S_D \cdot C \cdot \frac{D}{x} \tag{15}$$

The integral of Equation (15) is as follows:

$$\int_0^x x dx = \int_0^t \frac{C \cdot D}{\rho_{NbSi_2}} dt \tag{16}$$

Therefore,

$$\frac{1}{2}x^2 = \frac{C \cdot D}{\rho_{NbSi_2}} t \tag{17}$$

According to Equations (7) and (17),

$$(\Delta W_2)^2 = \frac{2S_D{}^2 \cdot \left[\left(a\rho_{Nb_5Si_3} - \rho_{NbSi_2}\right)\right]^2 C \cdot D}{\rho_{NbSi_2}} t \tag{18}$$

where $K_k = \frac{2S_D{}^2 \cdot \left[\left(a\rho_{Nb_5Si_3} - \rho_{NbSi_2}\right)\right]^2 C \cdot D}{\rho_{NbSi_2}}$.

In this equation, $\Delta W_2$ is the mass change during diffusion.

The kinetics model of the second stage of oxidation is

$$(\Delta W_2)^2 = K_k \cdot t \tag{19}$$

Equation (19) indicates that in the second oxidation stage of $NbSi_2$, the relationship between oxidation weight gain and oxidation time is in accordance with the parabolic rate law.

(3)    Third Stage of Oxidation

The shedding stage of the constant-temperature test was accompanied by oxidation and diffusion. The shedding of the sample only changed the quality of the sample, and did not change the mechanism of oxidation and diffusion of the sample. The complex process and the basic process with different oxidation degrees and contributions, according to the result of the analysis of stages of oxidation and diffusion dynamics, fell off the stage of oxidation caused by the oxidation rate and constant-temperature test and showed a linear relationship with the rate of oxidation stage. Diffusion caused by the oxidation rate and constant-temperature test showed a linear relationship with the rate of the diffusion phase.

The sample mass increases to $W_3$ at moment $t_{t1}$, and the sample mass increases to $W_4$ at moment $t_{t2}$. In order to accurately obtain the influence of oxidation and diffusion on the shedding stage, it is assumed that the shedding mass value per unit of time in the shedding stage is fixed, and that the mass change caused by shedding and oxidation occurs at the initial moment in the period. At moments $t_{t1}$–$t_{t2}$, the mass dropped from the sample is $W_5$, and the mass increased due to oxidation is $W_6$. Therefore, the oxidation rate $K_{DT}$ resulting from oxidation in the shedding stage is:

$$K_{DT} = mK_y, 0 \le m \le 1 \tag{20}$$

The oxidation rate $K_{KT}$ caused by diffusion in the shedding stage is:

$$K_{KT} = nK_k, 0 \le n \le 1 \tag{21}$$

Then, the actual mass increase $W_7$ of the sample at moment $t_{t1}$ is:

$$W_7 = W_3 + W_5 - W_6 \tag{22}$$

Since the mechanism of oxidation and diffusion of the sample is not changed at the shedding stage, according to Wagner and Daniel Monceau's oxidation theory the mass change of the sample at the shedding stage caused by diffusion is in accordance with:

$$W_4^2 - W_7^2 = nK_k(t_{t2} - t_{t1}) \tag{23}$$

According to Equations (21)–(23), it can be concluded that:

$$W_4^2 - (W_3 - W_6 + W_5)^2 = nK_k(t_{t2} - t_{t1}) \tag{24}$$

$$(W_4)^2 = m^2K_y^2(t_{t2} - t_{t1})^2 + [nK_K - 2mK_y(W_3 + W_5)](t_{t2} - t_{t1}) + (W_3 + W_5)^2 \tag{25}$$

Therefore, the physical form of the shedding stage dynamic model is:

$$(\Delta W_t)^2 = K_{T1}t^2 + K_{T2}t + c \tag{26}$$

In summary, the oxidation kinetics model of silicide coatings under isothermal condition in different stages is as follows.

$$\text{kinetics of oxidation} \begin{cases} \text{first stage}: \ \Delta W_y \ = K_Y t \\ \text{second stage}: (\Delta W_k)^2 = K_k t \\ \text{third stage}: (\Delta W_t)^2 = K_{T1}t^2 + K_{T2}t + c \end{cases} \tag{27}$$

## 5. Results and Analysis

The thermogravimetric experiment was carried out on the silicide coating using the developed thermogravimetric experimental device. The length, width, and height of the coating were $7 \times 10^{-2}$ m, $1 \times 10^{-2}$ m, and $1 \times 10^{-3}$ m, respectively. The bare surface area of the specimen was $0.156 \times 10^{-4}$ m$^2$. The weight was obtained online every ten min. The low-temperature experiments were carried out at $-180\,°C$ and $-100\,°C$. The results are shown in Figure 2. The room-temperature experiments were carried out at $60\,°C$, $80\,°C$, and $100\,°C$, the results of which are shown in Figure 3. The moderate-temperature experiments were carried out at $500\,°C$, $1000\,°C$, and $1700\,°C$. The results are shown in Figure 4. The high-temperature experiments were carried out at $2000\,°C$ and $2300\,°C$. The results are shown in Figure 5. The red lines in Figures 3–5 represent an oxidation kinetics numerical model which is summarized in Equation (27). Additionally, the main task of the experiments is to obtain proper coefficients for all oxidation stages in Equation (27).

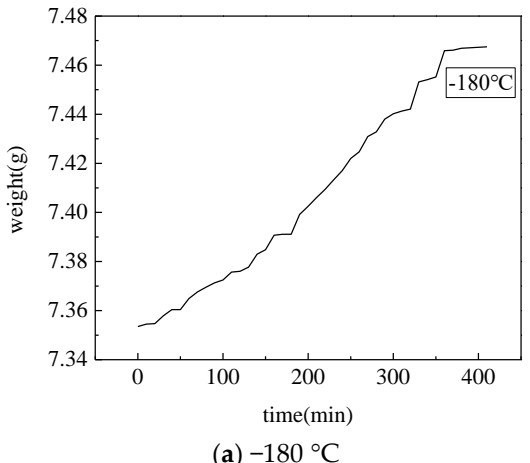

(**a**) −180 °C

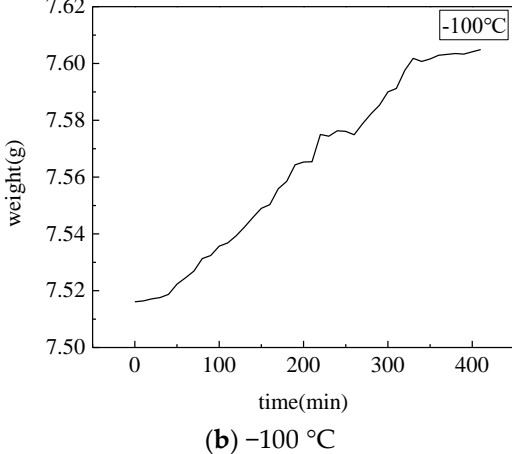

(**b**) −100 °C

**Figure 2.** Low-temperature experiment.

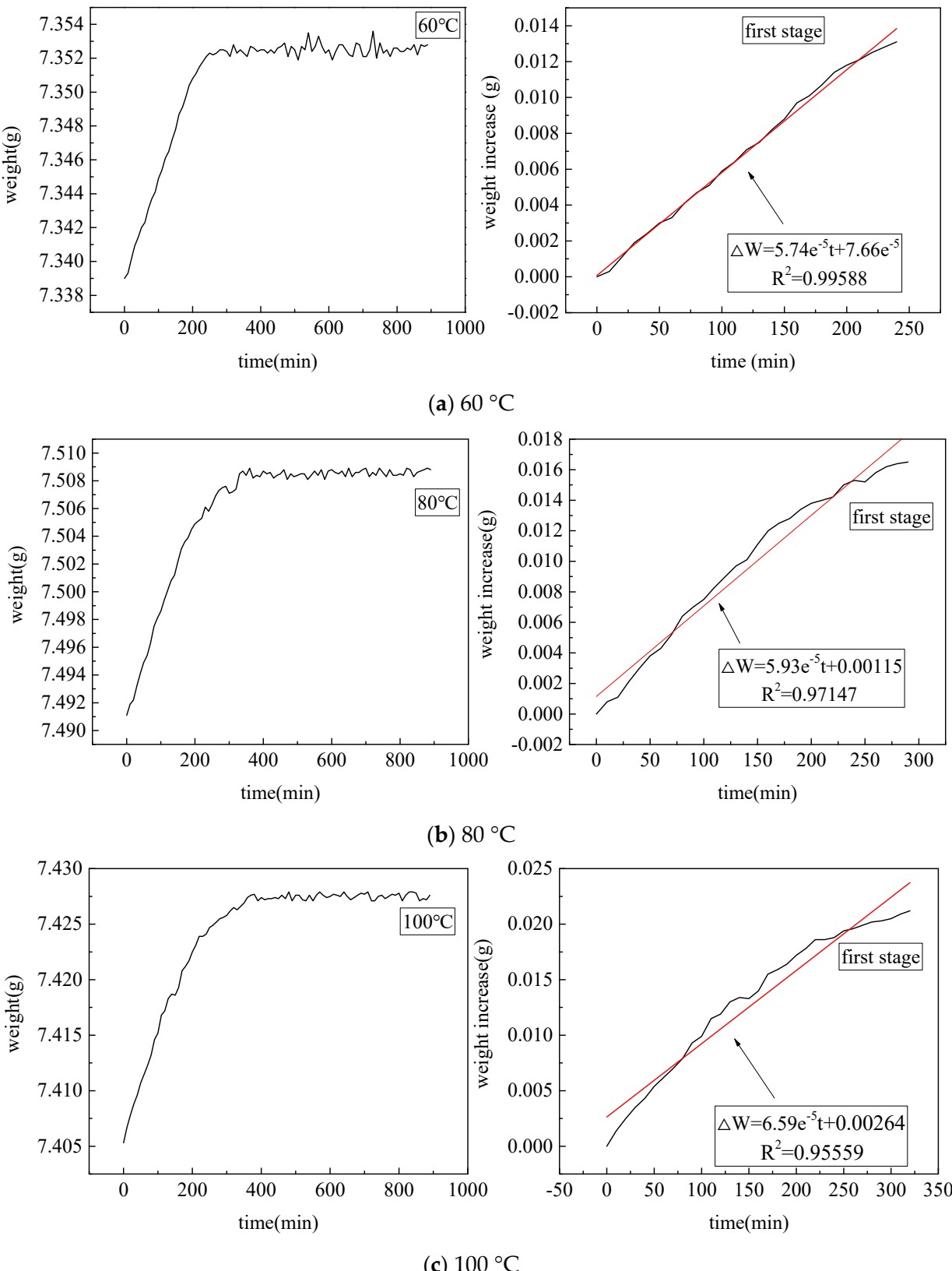

**Figure 3.** Room-temperature experiment.

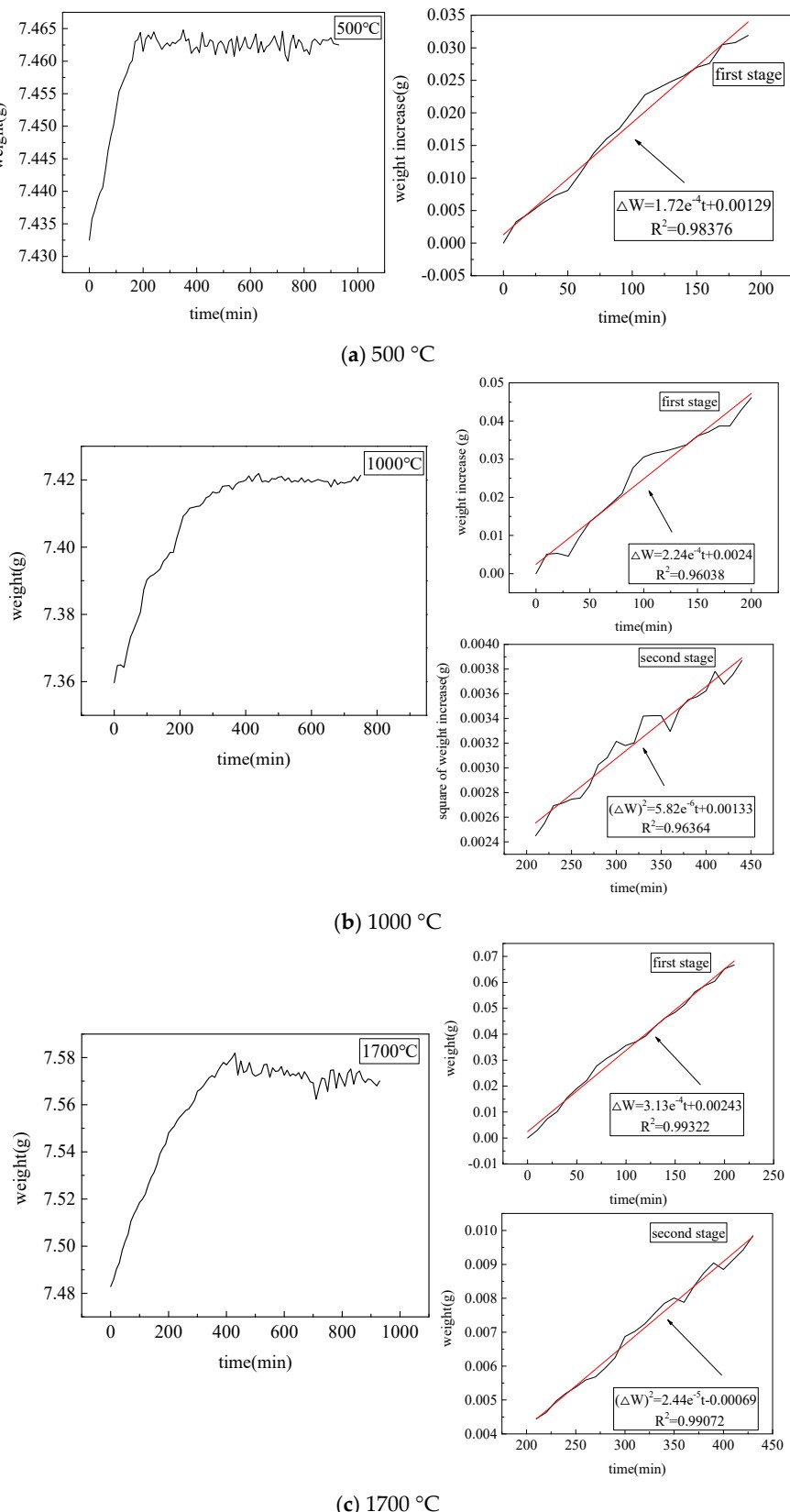

(**a**) 500 °C

(**b**) 1000 °C

(**c**) 1700 °C

**Figure 4.** Moderate-temperature experiment.

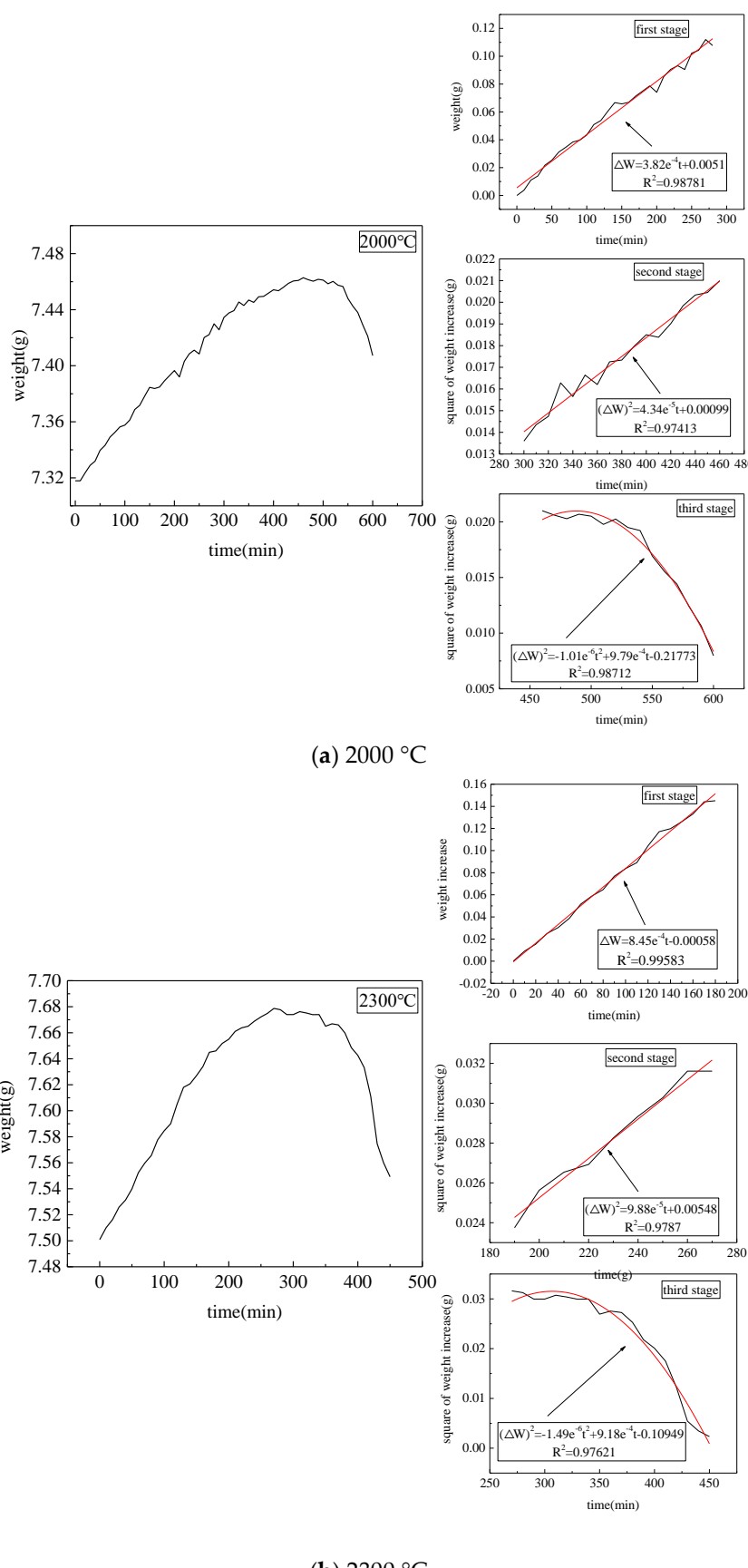

(**a**) 2000 °C

(**b**) 2300 °C

**Figure 5.** High-temperature experiment.

As shown in Figure 2, in the thermogravimetric experiments at $-180\ °C$ and $-100\ °C$, the weight gains of the specimen were 0.114 g and 0.0888 g, respectively. The weight gain and time were basically in a linear relationship. Thus, it can be inferred that the weight gain of the specimen was because of condensation or frost on the surface. In addition, the reason that the weight and time did not have a complete linear relationship was that low temperature might lead to cracks on the surface, which caused weight loss in the specimen.

In the thermogravimetric experiments carried out at $60\ °C$, $80\ °C$, and $100\ °C$ (as shown in Figure 3), the weight gains of the specimens were 0.0138 g, 0.0178 g, and 0.0223 g, respectively. According to Equations (1) and (2), the oxidation weight gains per unit were $0.0885 \times 10^4\ g/m^2$, $0.1139 \times 10^4\ g/m^2$, and $0.1429 \times 10^4\ g/m^2$. The average oxidation rates were 0.9939 g/(m$^2$·min), 1.2797 g/(m$^2$·min), and 1.6056 g/(m$^2$·min). Figure 3 shows that the weight of the specimen increased in the first place and then became stable gradually. This was because, in the first stage, the experimental temperature was quite low, so the oxidation only occurred on the surface. In addition, low temperature caused slow element diffusion. After oxidation, the surface of the coating became stable, and thus the specimen weight was finally stable. Based on the kinetics analysis method established in this paper, it was clear that when the specimen was in the linear stage of $60\ °C$, its oxidation rate was 0.0000574 g/(min), the correlation coefficient was $R^2 = 0.99588$, and then it became stable. When the specimen was in the linear stage of $80\ °C$, its oxidation rate was 0.0000593 g/(min), the correlation coefficient was $R^2 = 0.97147$, and then it became stable. When the specimen was in the linear stage of $100\ °C$, its oxidation rate was 0.0000659 g/(min), the correlation coefficient was $R^2 = 0.95559$, and then it became stable.

In the thermogravimetric experiments carried out at $500\ °C$, $1000\ °C$, and $1700\ °C$ (Figure 4), the weight gains of the specimens were 0.0300 g, 0.0617 g, and 0.0873 g, respectively. According to Equations (1) and (2), the oxidation weight gains per unit were $0.1920 \times 10^4\ g/m^2$, $0.3955 \times 10^4\ g/m^2$, and $0.5596 \times 10^4\ g/m^2$. The average oxidation rates were 2.0645 g/(m$^2$·min), 5.2700 g/(m$^2$·min), and 6.017 g/(m$^2$·min). In the thermogravimetric experiment carried out at $500\ °C$, the weight of the specimen increased in the first place and then gradually became stable. In the thermogravimetric experiments carried out at $1000\ °C$ and $1700\ °C$, the weight increased rapidly at first, and then rose slowly, finally becoming stable. In the thermogravimetric experiment carried out at $500\ °C$, an oxidation reaction occurred on the surface of the coating. The low temperature caused ineffective diffusion, so the surface of the coating became stable after oxidation. Hence, the weight of the specimens finally became stable. In the thermogravimetric experiments carried out at $1000\ °C$ and $1700\ °C$, an oxidation reaction occurred on the surface of the coating in the initial stage. After a period of time, diffusion occurred within an oxidation reaction. Oxidation, diffusion, and fall-off of the coating surface affected the weight loss of the coating. However, the low temperature and short experimental time caused the fall-off effect to be weaker than that of the others. Eventually, the weight of the specimen became stable. By analyzing the linear weight gain stage based on the kinetics analysis method established in this paper, it was clear that, when the specimen was in the first oxidation stage of $500\ °C$, its oxidation rate was 0.000172 g/(min), the correlation coefficient was $R^2 = 0.98376$, and then it became stable. When the specimen was in the first oxidation stage of $1000\ °C$, its oxidation rate was 0.000224 g/(min) and the correlation coefficient was $R^2 = 0.96038$. In the second stage of oxidation, the oxidation rate was 0.00000582 g/(min). The relevancy parameter was $R^2 = 0.96364$. Then, it became stable. When the specimen was in the first oxidation stage of $1700\ °C$, its oxidation speed was 0.000313 g/(min) and the correlation coefficient was $R^2 = 0.99322$. In the second stage of oxidation, the oxidation rate was 0.0000244 g/(min) and the correlation coefficient was $R^2 = 0.99072$. Then, it became stable.

In Figure 5, in the thermogravimetric experiments carried out at $2000\ °C$ and $2300\ °C$, the weight gains of the specimen were 0.0893 g and 0.0483 g, respectively. According to Equations (1) and (2), the oxidation weight gains per unit were $0.5720 \times 10^4\ g/m^2$ and $0.3096 \times 10^4\ g/m^2$, respectively. The average oxidation rates were 9.5330 g/(m$^2$·min) and

6.88 g/(m$^2$·min). In the thermogravimetric experiments carried out at 2000 °C and 2300 °C, the weight of the specimen was increasing rapidly at first, and then rising slowly, finally tended to decrease gradually. From analysis, it was quite clear that, in the initial stage, the fast oxidation on the surface led to a rapid increase of weight. After a period of time, diffusion occurred within an oxidation reaction, causing a slow tendency of weight gain. Finally, the weight declined because of the fall-off effect. Analyzing the linear weight gain stage based on the established kinetics analysis method, when the specimen was in the first oxidation stage of 2000 °C, its oxidation rate was 0.000382 g/(min) and the correlation coefficient was R$^2$ = 0.98781. In the second stage of oxidation, the oxidation rate was 0.000434 g/(min) and the correlation coefficient was R$^2$ = 0.97413. The kinetics model of the third stage of oxidation was $(\Delta W_3)^2 = -1.01e^{-6}t^2 + 9.79t - 0.21773$ and the correlation coefficient was R$^2$ = 0.98712. When the specimen was in the first oxidation stage of 2300 °C, its oxidation rate was 0.000845 g/(min) and the correlation coefficient was R$^2$ = 0.99583. In the second stage of oxidation, the oxidation rate was 0.0000988 g/(min) and the correlation coefficient was R$^2$ = 0.9787. The kinetics model of the third stage of oxidation was $(\Delta W_3)^2 = -1.49e^{-6}t^2 + 9.18e^{-4}t - 0.10949$ and the correlation coefficient was R$^2$ = 0.97621.

## 6. Conclusions

1. To achieve the online isothermal thermogravimetric experiment of silicide coatings, an online apparatus for isothermal thermogravimetric measurements was developed based on thermogravimetric analysis. This paper has studied two measuring methods: the weight gain and weight loss measuring methods. Additionally, the experiment was carried out using silicide coatings under a physical oxidation process carried out at between −180 °C and 2300 °C.

2. Based on the analysis of oxidation kinetics, this paper analyzes the internal mechanisms and kinetic parameters of the three stages of silicide coating oxidation, diffusion, and shedding, and derives their subsequent oxidation kinetic characteristics. On the basis of mathematical physics methods, a kinetics model of silicide coatings in different stages of oxidation was established, including parameters such as weight change, oxidation rate, oxidation time, etc. The results proved the perfect consistency of the kinetics model proposed by this paper and the oxidation process of silicide coatings.

3. In this paper, a complete kinetics model including different oxidation stages was firstly proposed for the whole oxidation process of silicide coatings, and the oxidation mechanism was revealed. This paper will play a significant role in the study of preparation technology enhancement and high-temperature environment applications, which also offers a theoretical foundation for accelerated aging and life evaluation methods. Based on the research in this paper, the focus of further research will be the study of the temperature corresponding to time mass changes.

**Author Contributions:** All authors contributed equally to the reported research. Conceptualization, D.A.; formal analysis, project administration, J.D. and P.X.; data curation, D.S., J.Z., Z.L., Y.X. and M.G. All authors have read and agreed to the published version of the manuscript.

**Funding:** This research is supported by National Natural Science Foundation of China (Grant No. 52206225, 61575029), National Defense Technical Basic Research Program of China (Grant No. JSZL2015603B002), and aviation Science Fund Project (Grant No.20172777007).

**Institutional Review Board Statement:** Not applicable.

**Informed Consent Statement:** Not applicable.

**Data Availability Statement:** All data generated or analyzed during this study are included in this published article.

**Conflicts of Interest:** The authors declare no conflict of interest.

**Nomenclature**

| Symbol | Description | Unit |
|---|---|---|
| $m_{h1}$ | weight of the specimen before experiment | g |
| $m_{h2}$ | weight of the specimen after $t_{h1}$ min | g |
| $S_{h1}$ | original surface area | $m^2$ |
| $G_h^+$ | oxidation weight gain per unit | $g/m^2$ |
| $\overline{K_h^+}$ | average oxidation rate | $g/m^2$ min |
| $m_{h3}$ | weight of the specimen before experiment | g |
| $m_{h4}$ | weight of the specimen after $t_{h2}$ min | g |
| $S_{h2}$ | original surface area | $m^2$ |
| $G_h^-$ | oxidation weight loss per unit | $g/m^2$ |
| $\overline{K_h^-}$ | average oxidation rate | $g/m^2$ min |
| $S_D$ | sectional area of the coating | $cm^2$ |
| $\rho_{NbSi_2}$ | density of $NbSi_2$ | $g/cm^3$ |
| $\rho_{Nb_5Si_3}$ | density of $Nb_5Si_3$ | $g/cm^3$ |
| $x$ | oxidation thickness of $NbSi_2$ | cm |
| $W0$ | mass of $NbSi_2$ with thickness $x$ | g |
| $W1$ | mass of the sample when $Nb_5Si_3$ is completely oxidized | g |
| $\Delta W$ | change of the oxidation mass of the sample | g |
| $\alpha$ | thickness conversion factor | |
| $v_{D1}$ | oxidation rate of $NbSi_2$ | cm/s |
| $v_h$ | chemical reaction rate | cm/s |
| $k_1$ | oxidation rate constant | |
| $C$ | oxygen concentration in the air | |
| $\Delta W1$ | mass change during oxidation | g |
| $\Delta W2$ | mass change during diffusion | g |

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
