# Peer review of "Analysis and Kinetics Modeling of the Isothermal Oxidation Behavior of Silicide Coatings"

_coatings, doi:10.3390/coatings13081464_

Round 1

Reviewer 1 Report

Authors propose a disigned apparatus for online thermogravimetric measurements (TGA) in a wide temperature range that is of high engineering and practicle value for everyone engaged into resistant coating technology. However, one can find that mentioned apparatus has been already developed in 2019. Moreover, Nb-Hf alloy was investigated as well. The fact that Si–Cr–Ti silicide coating was applied instead of currently used NbSi2 is insuffitient to be considered as the new findings to publish (see Metrol. Meas. Syst., Vol. 26 (2019) No. 4, pp. 725–737 DOI: 10.24425/mms.2019.130563). In addition, an importance and necessity of the low temperature TGA is rather questionable. There are no articles and reports dealing with TGA measurements at the temperature of liquid nitrogen and below excepting some superconducting ones that is obviously out of focus in this case (Niobium hafnium alloy has high melting point, fine anti-corrosion and workability, it can be used for pharmacy, semiconductor, aviation, nuclear, etc.). As far as thermogravimetric measurements are concerned, there are several critical issues that must be discussed prior publication it anywhere. First, what about oxidation kinetics of the bare Nb-Hf alloy? How it is possible to distinguish between Nb-Hf and Nb-Hf/NbSi2 oxidation processes without any reference sample? Secondly, authors didn't provide any kinetics model of the oxidation process, its just a simple pure empirical observations and raw fitting of the results obtained. The drawback of the report presented is an absent of any microstructural characteristics (XRD, SEM) and chemical composition analysis (EDX etc.) confirming oxidation. For instance, it is not clear at which exact temperature the NbSi2 oxidation starts. In addition, why authors say that a product of oxidation is Nb5Si3 instead of a simple reaction: 4NbSi2+13O2=8SiO2+2Nb2O5 (of course that a final product can contain Nb5Si3 as well, but most of the authors claimed that its fraction is rather small). The initial characteristics of the silicide coated Nb-Hf specimen is not provided as well (porosity, microstructure, thickness etc.). Finally, obtained weight increase dependencies on time under a fixed oxidation temperature are not discussed at all, nothing to say about any comparisson with the similar previously obtained results (https://doi.org/10.1016/j.corsci.2017.10.002 for example). 

That is why, my overall recomendation is to reject current article due to some plagiarism detected, absent of enough scientific novelty and rather floopy experimental part conducted.

The overall English style is very simplified even compared with detected plagiarism at Metrol. Meas. Syst., Vol. 26 (2019) No. 4, pp. 725–737 DOI: 10.24425/mms.2019.130563. The English is understandable but very barren and contains monotypic both lexis and grammar. 

Reviewer 2 Report

Authors carried out theoretical investigations on oxidation kinetic mechanisms in order to develop a kinetics model of silicide coatings, which was then experimentally validated. The manuscript contains significant accomplishments that will be beneficial to the readers. Therefore, I recommend publishing this manuscript in ‘Coatings Journal’. Following minor comments need to be addresed;

1. Abstract first line - provide the problem statement / need for this study for better understanding

2. Add sufficient literature in the introduction part for oxidation and failure of silicide coatings to understand the research gap.

3. In section 2, the experimental details of silicide coatings oxidation process are not clear.

4. Experimental results need to be correlated with existing literature evidences in the results and discussion section.

5. English should be polished.

Reviewer 3 Report

The paper is interesting and needs only some points for improvement.

- please position your equations in a structured way. They are somehow 'flying around' in the text.

p.2,

- line 51: 'model, which'

- line 74: '0.1 mg'

You have 2 times a chapter 3. Please, renumber your chapters.

Chapter 'Oxidation kinetics...': at the beginning, you mention 4 steps of oxidation behaviour, later on you speak about 3 stages of oxidation reaction. Please, stick to one definition of steps/stages.

p.7, eq. 22: W5 is the mass loss due to shedding, W6 is the gain in mass by oxidation. Why do you add the mass loss W5 in eq. 22, while the mass gain W6 is with a '-' to calculate the mass gain W7?

p.7, line 3 from the bottom: 'results are shown in'

Reviewer 4 Report

1) In the Abstract the Authors should add more the most important obtained results with an aim to highlight the most important elements obtained in the research. At the moment, the most important findings are only generally mentioned in the Abstract – exact elements should be added and highlighted.

2) Section 2 - Thermogravimetric Experimental Apparatus – the Authors should present at least a general specifications and accuracy/precision of each used measuring device. If possible, the Authors should also present overall accuracy/precision range for each measured parameter (or at least a discussion related to the overall accuracy/precision range for each measured parameter should be added).

3) It seems that the paper is written in a hurry, without proper check and corrections before its upload to the Journal website. First of all, throughout the paper can be found many obvious and typing mistakes. The equations are not properly arranged (they should stand in the middle of the paper). In the paper occur two Section 3 – page 4. Many other similar mistakes are obvious in the paper – during the paper reading they become really annoying after some time. Therefore, a careful and proper paper arrangement and correcting all obvious and typing mistakes is highly required.

4) Due to numerous abbreviations, symbols and markings used throughout the paper, it is advisable that the Authors add in the paper a Nomenclature inside which will be listed and explained all abbreviations, symbols and markings used throughout the paper text. A Nomenclature will notably improve paper reading experience.

5) Figures 3, 4 and 5 – red lines represent an oxidation kinetics numerical model (summarized in the equation 27). If I understand correct – the main aim of the performed experiments is to obtain proper coefficients for all oxidation stages in equation 27. However, proper explanations related to these red lines are required, it is not fully understandable how are they obtained and what is the main task (these elements are missing in the explanations of Figures 3, 4 and 5).

6) In the Conclusions section should be added guidelines in further research related to this topic and the possibilities of obtained results further usage (exact possibilities, not only general recommendations).

7) English is readable and understandable, but it should be improved in many sentences or whole paper parts. Please, perform careful checks and improvements related to the English.

8) The List of References is not proper. The Authors should add recent literature in the List of References because almost all references are older than 10 years. I can understand that for some basic literature is irrelevant how old it is, but at the moment the List of References did not prove that performed research is scientifically relevant, actually it confirms that researches in this field were important in the past, but not nowadays. Therefore, a notable improvement related to the used Literature and adding recent literature form this research field is highly required.

Final remarks: This is an interesting research, but it should be notably improved, corrected and upgraded (according to the comments above) to be a proper publication candidate.

English is readable and understandable, but it should be improved in many sentences or whole paper parts. Please, perform careful checks and improvements related to the English.

Round 2

Reviewer 1 Report

Among all the suggestions I've made, the only things that author had  work on is English correction. In the current form manuscript is well readable. Concerning other criticism, authors did not show any improvements. That is why, I am solid for a reject decision for the revised manuscript.

1) Authors responded that "Instead, this device covers low to high temperatures, has not been developed before, and the service environment needs to be simulated from low to high temperatures." However, the current apparatus covesr -180 to 2300 degrees temperature range, while prior apparatus (described at Metrol. Meas. Syst., Vol. 26 (2019) No. 4, pp. 725–737) does from -180 degrees too, but provided even higher maximal temperature of 3000 C.  Authors claims that the current apparatus can simulate cryogenic enviroment, however it is obvious that -180C is a liquid nitrogen temperature range rather than cryogenic. That is why, presented aparatus does not provide any new feature and possibility compared with published in Metrol. Meas. Syst. by authors. Authors even did not mention and cite their own research throwing a dust in readers eyes that is not ethical at all.

2) Concerning other responses (#2-#9), I must to point out that the journal title is "Coatings" rather than "Apparatus". That is why, it assumes coatings investigations in accordance with well known protocols and experimental design, which include SEM, XRD etc. If one check the current title, it includes "Kinetics Modeling", however authors clearly responded that "The purpose of this article written by the author is to study the overall performance of coatings on Nb-Hf alloys without studying Nb-Hf alloys. Therefore, no distinction is made in this paper between the oxidation kinetics of this alloy"

In summary, it is obviously an exactly one and the same apparatus described in 2019 and the current manuscript does not specify any valuable properties of the Hf-Nb alloyed coating.

Reviewer 4 Report

In the revised paper occur some notable confusion, or the Authors uploaded wrong paper version. The Authors have answered to my comments, but the most of the answers WERE NOT implemented in the paper. I will refer to my comments related to the previous paper version:

COMMENT 1) In the Abstract the Authors should add more the most important obtained results with an aim to highlight the most important elements obtained in the research. At the moment, the most important findings are only generally mentioned in the Abstract – exact elements should be added and highlighted.

REVISED PAPER VERSION: The Authors have performed some improvements, but still I don't see that the Abstract contains the most important findings from the research, it is still general and overall.

COMMENT 2) Section 2 - Thermogravimetric Experimental Apparatus – the Authors should present at least a general specifications and accuracy/precision of each used measuring device. If possible, the Authors should also present overall accuracy/precision range for each measured parameter (or at least a discussion related to the overall accuracy/precision range for each measured parameter should be added).

REVISED PAPER VERSION: Table 3-1 and 3-2 are placed in the answers to my comments, but they are not involved in the paper. They should be added in the paper at least in the Appendix.

COMMENT 4) Due to numerous abbreviations, symbols and markings used throughout the paper, it is advisable that the Authors add in the paper a Nomenclature inside which will be listed and explained all abbreviations, symbols and markings used throughout the paper text. A Nomenclature will notably improve paper reading experience.

REVISED PAPER VERSION: A Nomenclature is NOT added in the paper (regardless of the fact that in the answers to my comments the Authors have stated: "the author added  a Nomenclature").

COMMENT 6) In the Conclusions section should be added guidelines in further research related to this topic and the possibilities of obtained results further usage (exact possibilities, not only general recommendations).

REVISED PAPER VERSION:  The Conclusion should have some kind of structure, not only the taxative highlights. Further research and the possibilities of obtained results further usage is not properly explained and elaborated (only one general sentence is surely not proper explanation).

COMMENT 8) The List of References is not proper. The Authors should add recent literature in the List of References because almost all references are older than 10 years. I can understand that for some basic literature is irrelevant how old it is, but at the moment the List of References did not prove that performed research is scientifically relevant, actually it confirms that researches in this field were important in the past, but not nowadays. Therefore, a notable improvement related to the used Literature and adding recent literature form this research field is highly required.

REVISED PAPER VERSION:  The new literature is added, but again, the dominant amount of that literature is not recent. I believe that the further improvement can be performed.

FINAL REMARKS: The Authors should perform and implement all the elements mentioned in my first review inside the paper. At the moment - the answers to my comments and corrections performed in the paper are not consistent. Therefore, I believe that the further improvement of this paper is required.

Round 3

Reviewer 1 Report

There are no comments to be adressed anymore

Author Response

Thank you to the reviewer for your careful review.